# Influence of Design Parameters of Idler Bearing Units on the Energy Consumption of a Belt Conveyor

**Piotr Kulinowski, Piotr Kasza *** and **Jacek Zarzycki**

Department of Machinery Engineering and Transport, Faculty of Mechanical Engineering and Robotics, AGH University of Science and Technology, 30-059 Cracow, Poland; piotr.kulinowski@agh.edu.pl (P.K.); jacek.zarzycki@agh.edu.pl (J.Z.)

* Correspondence: piotr.kasza@agh.edu.pl; Tel.: +48-502-755-831

**Abstract:** This publication presents the results of laboratory tests of idler rolling resistance under operational loads. Operational loads are understood as radial and axial forces acting on the idler, with values corresponding to those that occur in the conditions of its operation in copper ore mines. Knowing the rolling resistance is important not only at the stage of conveyor design, selection of the drive power or calculations of the necessary belt strength, but also when improving and searching for new idler design solutions. The idlers adopted for this research were differentiated in terms of bearings and idler axial clearance. The investigations were carried out on a unique test stand designed and built by the authors. The construction of the stand enables simulating operational loads while measuring the rolling resistance. The test rig measures idler bearing losses and rolling drag, not belt indentation rolling resistance. The object of the research were ø133×465 idlers, which are most commonly used in the raw materials industry. The results show the possibility of reducing the belt conveyor energy consumption by appropriate selection of the design features of the idler bearing unit.

**Keywords:** belt conveyor; carrying idler; rolling resistance; energy consumption; energy saving solutions

## 1. Introduction

The role of idlers in idler sets is to support, guide and form the belt along the conveyor route. Idlers are one of the most numerous components of the conveyor—there may be over 3000 idlers per one kilometer of the conveyor length. Therefore, despite the simple structure and relatively low unit price, idlers are an important element of belt conveyor operating costs. Thus, the energy consumption of the conveyor and its reliability largely depend on the quality of the idlers [1,2].

The parameter that allows for assessing the quality of the idler structure and performance is, among others, its rolling resistance. Idler rolling resistance is the value of the force that must be applied to the casing surface in order to overcome the friction resistance moment in the bearing units [3]. The value of idler rolling resistance is influenced by many factors related to its design, quality of workmanship and operating conditions [4–7]. Table 1 lists the sources of idler rolling resistance.

During operation, idlers in belt conveyors are loaded with radial and axial forces. The radial load comes from the weight and force of the belt tension as well as the weight of the material transported on the belt [8–10], whereas axial forces are generated in the case of idlers inclined or skewed in relation to the direction of belt movement. These loads cause stresses and deformations in the structural elements of the idler, which translate into an increase in its rolling resistance.

Permissible values of idler rolling resistance are specified in relevant standards, which recommend testing idler rolling resistance under a radial load of 250 N. However, the operating loads of the idler are much higher [11–13]. Therefore, standard requirements

should be treated as standardized guidelines for idler quality control during the production process.

**Table 1.** Sources of roller rotational resistance.

| Structural Factors | Manufacturing Factors | Exploitation Factors |
| --- | --- | --- |
| length | | belt speed |
| axis diameter | | load |
| roller diameter | machining accuracy | temperature |
| coat thickness | assembly accuracy | humidity and water |
| bearing quality | roller unbalance | dust and dirt |
| bearing clearance | inaccuracy in positioning the bearings | corrosion |
| type of sealing | matching of parts | location |
| type of grease | | operating times |
| fit of components | | human factor |

The testing of idlers with radial loads up to 8 kN have been presented in publications [5,10]. The applied measuring system, as well as mathematical functions, allowed for the theoretical description of the correlation between the rolling resistance of the tested idlers and the value of the applied radial force. Idlers operating in opencast mines have been tested. The tests described in the literature present the results of idler rolling resistance tests according to the recommendations known from the standards or take into account the radial load corresponding to the operational load. However, there are no studies on intentionally introduced axial loads, which undoubtedly occur during normal operation and have an impact on the idler rolling resistance. In one study [8], the value of the side idlers' axial load was estimated using simulation tests.

In the literature there is no clear information about the impact of operational loads on idler rolling resistance. The authors of this study also set the goal of determining the impact of idler structure on the rolling resistance. These were the reasons for the authors to undertake research into the influence of radial and axial loads as well as idler design parameters on rolling resistance.

## 2. Test Rig Description

Idler rolling resistance under load with radial and axial forces was tested on a special stand developed by the authors of this study. The general view of the test rig is shown in Figure 1. The stand is dedicated to testing idler rolling resistance under load, especially with radial operational load up to 5 kN and axial load up to 0.5 kN. The stand enables the testing of idlers with the casing diameters ranging from 108 to 259 mm and the casing length up to 1200 mm. The construction of the stand also allows for tilting the frame so that it is possible to reproduce the work of the trough roller inclined to the angle of 45°.

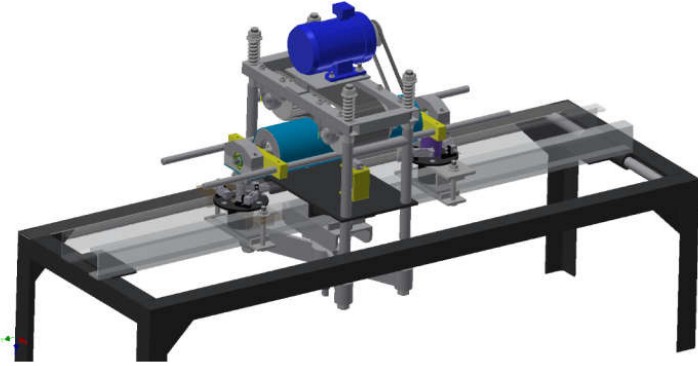

**Figure 1.** The general view of the test rig.

The test rig only applies to testing individual idler rolls, and not 2, 3 or 5 idler roll sets commonly used to support the carrying trough of actual belt conveyors. Since the test rig measures axial load resistance, its application to flat idler rolls in 35 degree (or any other angled trough) troughed sets will need to be considered and adjusted in the results, so that a designer may use the data in a real situation.

The photo in Figure 2 shows a fragment of the test stand with a visible idler and a system for driving and exerting a load with a toothed belt.

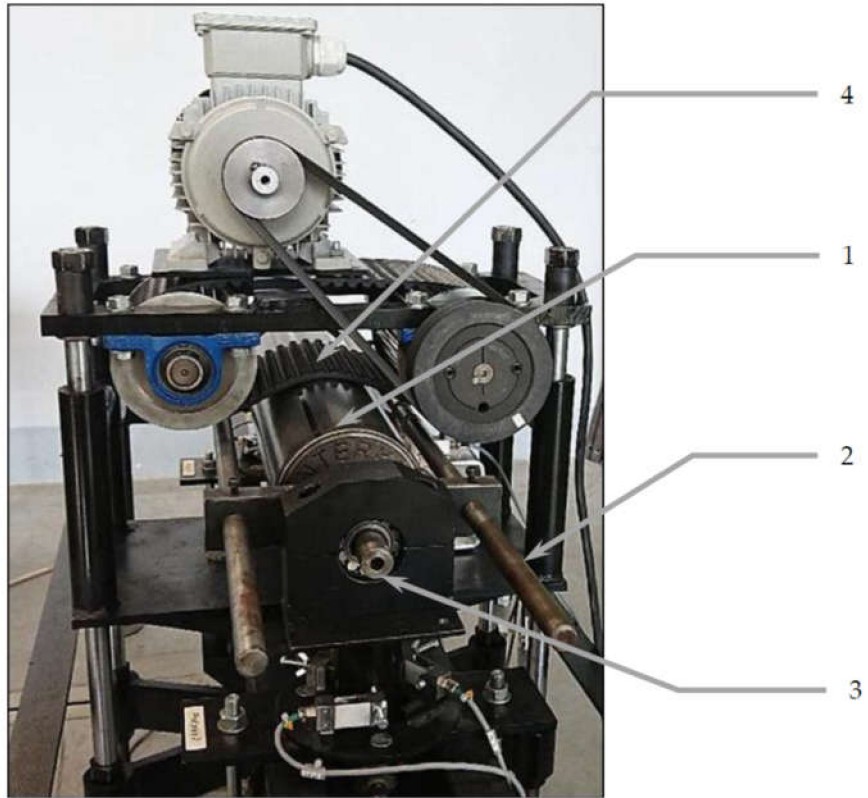

**Figure 2.** View of an idler mounted on a test stand (markings in the text).

The idea behind the measurement is that the tested idler (1) is mounted in a rigid frame (2), which is pivotally supported (3). The drive and setting of the radial load are effected by means of a toothed belt with high strength (4).

The diagram showing the method of exerting loads and determining the forces is presented in Figure 3. The axial load is applied through the contact of the idler with the hub in the idler axis (Figure 3b). During measurements, the reaction force $F$ acting on the frame on the arm $e$ is recorded as well as the radial force $Q$ loading the idler as the sum of forces in both supports $P_1$ and $P_2$. Idler rolling resistance $W$ is calculated from dependence (1):

$$W = \frac{F \cdot d}{2 \cdot e} [N] \tag{1}$$

where $F$ is the force measured with a force sensor, $d$ is the idler diameter and $e$ is the arm of force $F$.

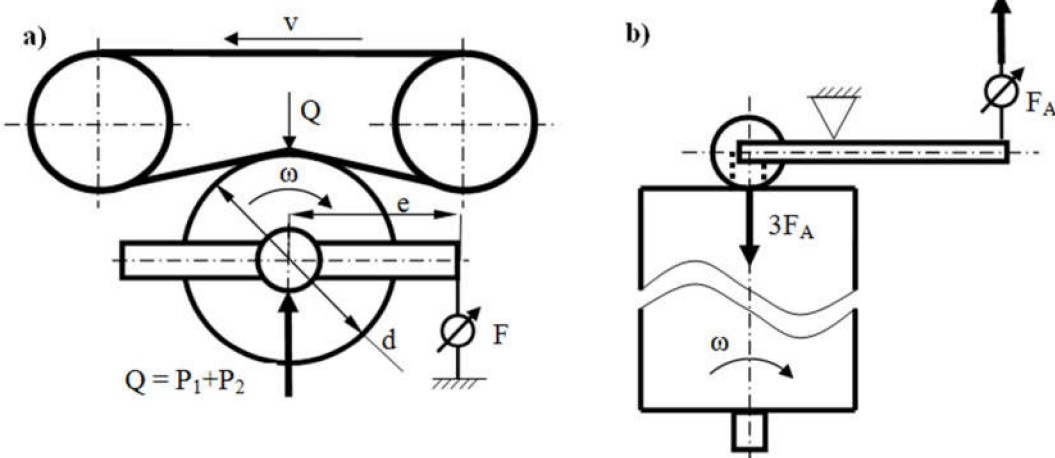

**Figure 3.** Diagram of a system for exerting radial load (**a**), axial load (**b**).

Force $F$ is measured with a force sensor within a measuring range of 100 [N] and a cumulative error of $\leq 0.20\%$. The frequency of recording force $F$ was 1 [kHz] and the measurement precision $\pm 0.13$ [N].

The axial force $F_A$ is exerted by means of a lever mechanism with a pressure roller. Owing to this solution, the axial force applied to the idler hub does not disturb the measurement of the idler rolling resistance force. This system is shown in Figure 3b. The leverage system with a ratio of 3:1 was equipped with a spring dynamometer, which enabled controlling the applied axial force.

### 3. Course and the Results of the Tests

The following independent variables were adopted for the testing of idler rolling resistance:

- radial force value of 1000 N,
- axial force value of 0–300 N,
- idler rotational speed of 650 1/min,
- left and right direction of idler revolution.

The values of the above parameters were selected so as to reflect the operational load affecting the idler when it is working on a real conveyor.

All the idlers were run-in in accordance with the standard requirements. Before the tests, each idler rotated for a minimum of 30 min in order to warm up the bearing units, which guaranteed repeatability of the results.

For each idler, tests of its rolling resistance were carried out in two series:

- series 1: radial force of 1000 N, axial force of 300 N, right revolutions,
- series 2: radial force of 1000 N, axial force of 300 N, left revolutions.

Each series consisted of 10 trials for each idler. The trial was divided into 5 time sequences lasting a total of 40 s. At the beginning, data recording was started; then, after 5 s, the idler drive was turned on and for another 10 s the rolling resistance was recorded only under the radial force load. From the 15thsecond of the test, the roller causing the axial force was in contact with the idler, but no axial force was applied. The purpose of this was to check whether the roller rotation resistance disturbed the measurement of the idler rolling resistance, which was assessed during the measurement results analysis. Between the 20th and 30th second of the test, an axial force was applied in addition to the radial force. After 30 s, the axial force load was removed, and at the 35th second, the idler was stopped; after another 5 s, data recording was completed. The diagram of idler loading is shown in Figure 4.

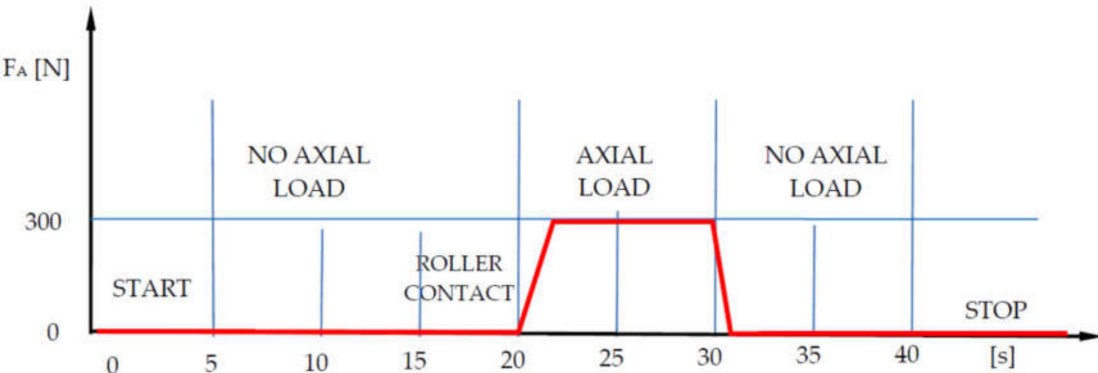

**Figure 4.** Diagram of loading the idler with axial force $F_A$.

The measurements enabled obtaining a record of the idler rolling resistance force over time, which was variable depending on the radial and axial force in both directions of the idler casing rotation.

The rolling resistance was determined on the basis of the average value calculated for 5 s of idler operation without and with axial force load. An example of changes in idler rolling resistance is shown in Figure 5.

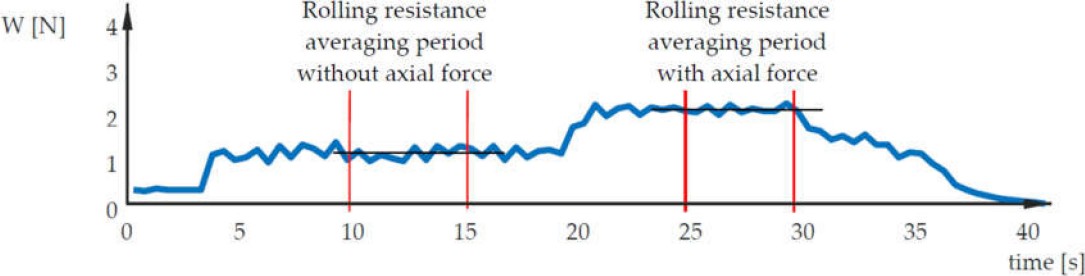

**Figure 5.** An example of the course of changes in idler rolling resistance during a single test.

The influence of changes in the axial force on the value of rolling resistance was assessed on the basis of the $k_{WX}$ coefficient, which was defined as the coefficient of change in rolling resistance per 100 N increment of the axial force. For example: with $k_{WX} = 1$, an increase in the axial force by 250 N will cause the resistance to increase by 2.5 N. The coefficient may have negative values. The course of the change in the value of the $k_{WX}$ coefficient depending on the increase in the axial force is shown in Figure 6.

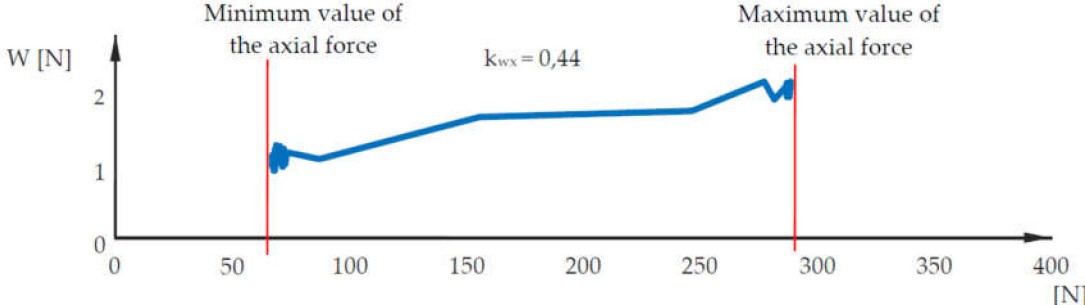

**Figure 6.** Sample record of the $k_{WX}$ coefficient value during a single test.

During the tests, the reaction forces acting on the idler supports in the direction of the X, Y and Z axes were recorded in accordance with the markings shown in Figure 7. The

reaction forces measured in both supports were derived from the radial and axial forces acting on the idler.

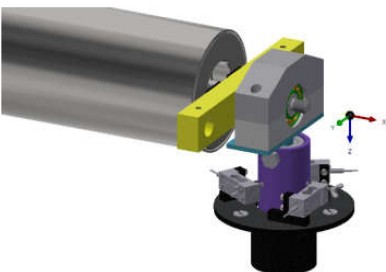

**Figure 7.** Marking of the direction of forces acting on the idler support.

The recorded courses of changes of the reaction forces $R_X$, $R_Y$ and $R_Z$ were used to determine the radial and axial forces acting on the idler casing. The course of changes in the radial and axial force during a single test is shown in Figure 8.

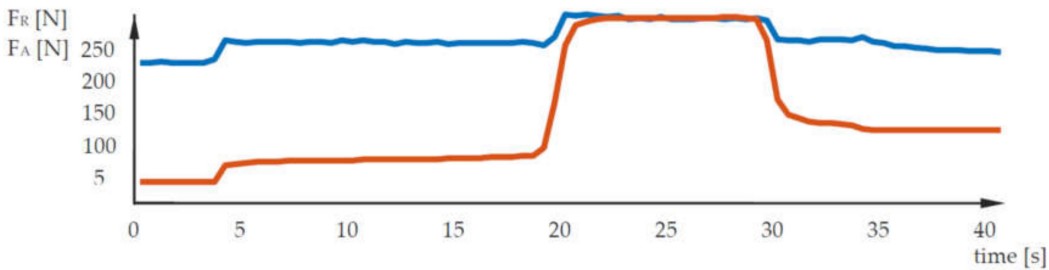

**Figure 8.** Sample record of the value of radial force $F_R$ (blue) and axial force $F_A$ (red).

The object of the research was a series of 12 ø133 × 465 idlers having a steel structure with a cast iron hub. The idlers differed from each other in the type and manufacturer of the bearing as well as the value of the axial clearance. In the tested idlers, 6305 ball bearings were used. The X-bearings were of low quality, while the Y-bearings were of high quality from a reputable manufacturer. Idlers 1 to 6 were equipped with bearings of C3 clearance class, and idlers 7 to 12with bearings of C4 clearance class. Three values of axial clearance L1, L2 and L3 were used in the tested idlers, where L1 < L2 < L3. Table 2 lists the markings with the parameters of the tested idlers.

**Table 2.** Markings and parameters of the tested idlers.

| Idler No. | Bearing Clearance | Bearing Producer | Idler Axial Clearance |
|:---:|:---:|:---:|:---:|
| 1 | | | L1 |
| 2 | | X | L2 |
| 3 | C3 | | L3 |
| 4 | | | L1 |
| 5 | | Y | L2 |
| 6 | | | L3 |
| 7 | | | L1 |
| 8 | | X | L2 |
| 9 | C4 | | L3 |
| 10 | | | L1 |
| 11 | | Y | L2 |
| 12 | | | L3 |

### 4. Discussion

As a result of the analysis of the data obtained during the performed measurements, average values of the rolling resistance for each idler were obtained. These were two values for each idler:

- rolling resistance under load with a radial force of 1000 N,
- rolling resistance under load with a radial force of 1000 N and an axial force of 300 N.

Based on the obtained results, an analysis of the influence of the load status as well as design parameters (X and Y bearings, C3 and C4 bearing clearance) and technological parameters (L1, L2 and L3 axial clearance) of idlers on their rolling resistance was performed.

The collective bar chart in Figure 9 shows two average values of rolling resistance for the 12tested idlers. Such presentation of the results enables assessing the impact of the idler load status on the rolling resistance.

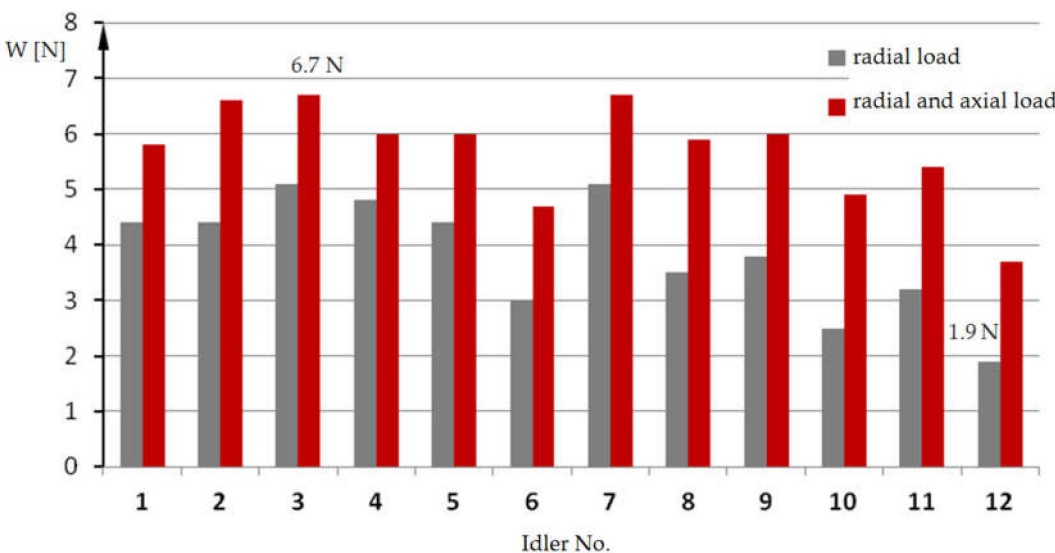

**Figure 9.** Results of measurement of idler rolling resistance under radial and radial + axial load.

The test results showed a significant impact of the load as well as the design and technological parameters of the idler on its rolling resistance. It can be seen in the diagram above that idlers of the same design may have a rolling resistance ranging from 1.9 to 6.7 N, depending on the type of bearing used.

Considering, for example, idlers 3 and 12, loaded only with a radial force, it can be seen that idler 3 has more than 2.6-times greater rolling resistance than idler 12. This means that idler rolling resistance can be considerably reduced without making significant changes in the structure and production technology of the idler only by using a high-quality bearing. The condition is a stable and repeatable production process, e.g., supervised by procedures and quality control tools.

A significant influence of the axial force on rolling resistance was observed for all the tested idlers. In the case of idlers 10 and 12, the value of rolling resistance almost doubled after loading with an axial force. The practical conclusion is that an increase in rolling resistance with axial force should be taken into account when designing belt conveyors.

In order to assess the impact of the idler's design and technological factors on its rolling resistance, the test results were compiled in a system that allows their comparison in terms of the feature being the subject of the analysis.

In order to compare the idler rolling resistance in terms of bearings produced by X and Y manufacturers, the tested idlers were compared in pairs, where they differed only in this feature. Namely, the idlers were arranged in pairs: 7-10, 8-11 and 9-12 as shown in the diagram in Figure 10.

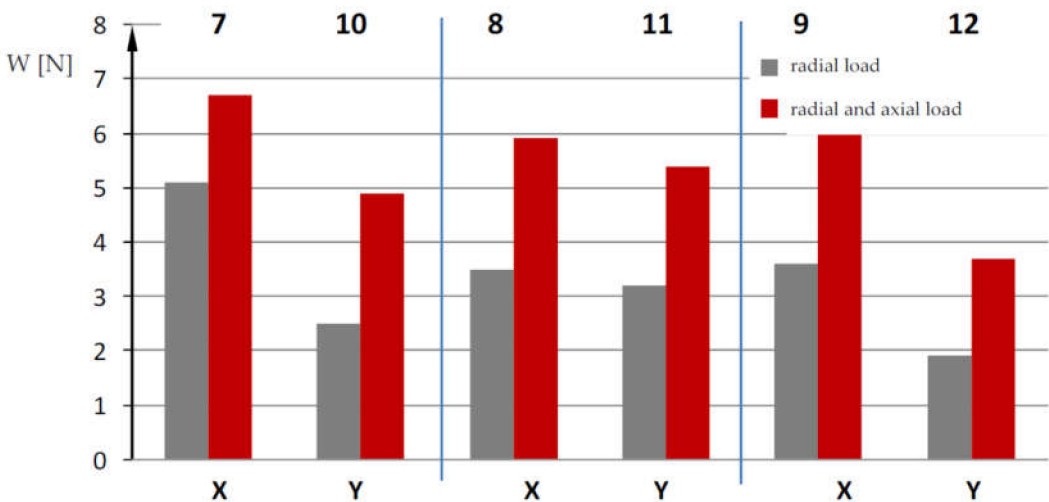

**Figure 10.** Comparison of idler rolling resistance in terms of the bearings used.

The results of the comparison have revealed that the use of Y bearings in idlers will reduce the rolling resistance. When analyzing the statistical parameters of the measurement data, it was also noticed that the waveforms of changes in rolling resistance during the tests were more stable and more repeatable for Y bearings.

In order to compare the bearing clearance of C3 and C4, the tested idlers were compared in pairs, where they differed only in this feature. Namely, idlers were arranged in pairs: 4-10, 5-11 and 6-12, as shown in the diagram in Figure 11.

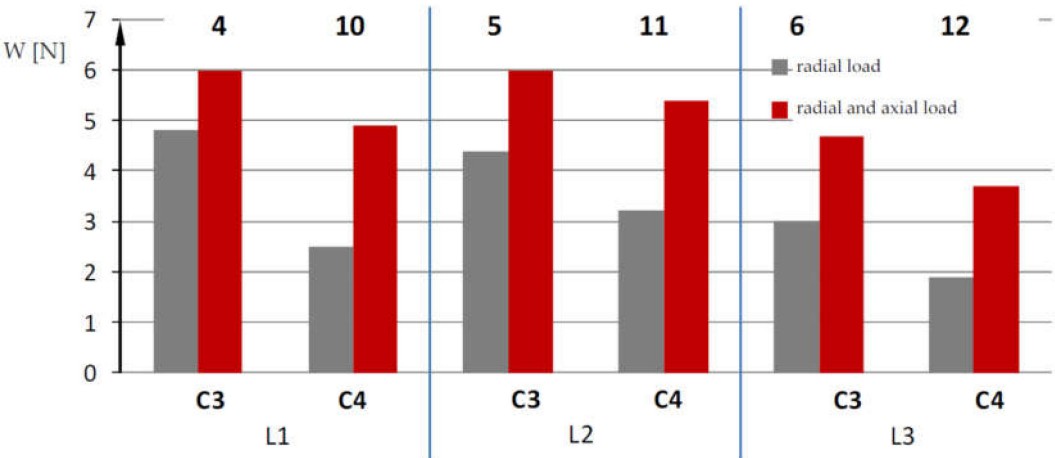

**Figure 11.** Comparison of idler rolling resistance in terms of C3–C4 bearing clearance—Y bearing.

In the case of Y bearings, a reduction in the idler rolling resistance can be observed whenever C4 bearing clearance is applied, compared to C3 clearance.

To compare idler rolling resistance with regard to axial clearance, the tested idlers from the same bearing manufacturer (X, Y), with the same clearance (C3, C4), but with different axial clearance (L1 < L2 < L3) were arranged in threesomes. Namely, idlers 4-5-6 and 10-11-12 were compared as shown in the diagram in Figure 12.

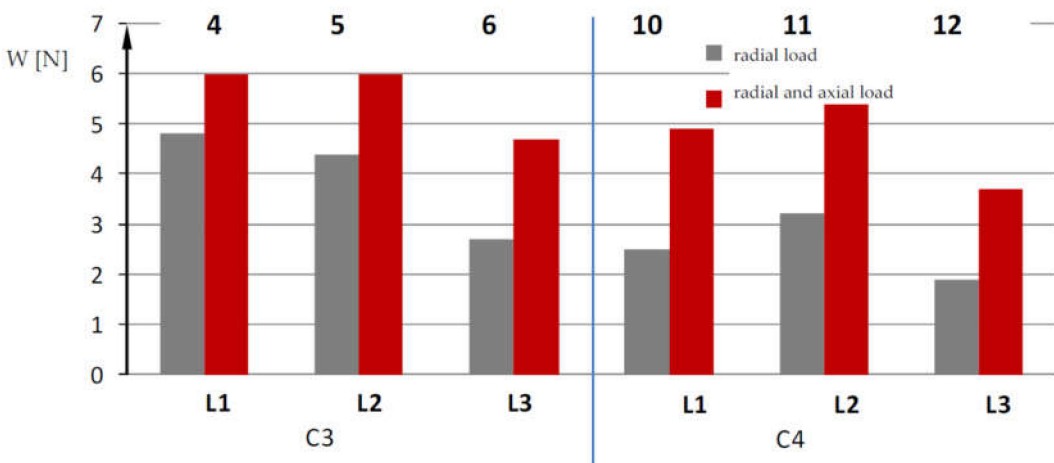

**Figure 12.** Comparison of idler rolling resistance in terms of axial play.

The analysis of the influence of axial clearances required during idler installation on its rolling resistance has revealed that the use of a correctly selected play L results in a reduction of the average value of rolling resistance.

In order to assess the energy effect that can be obtained after applying a certain type of idler in the conveyor, a computational example was prepared using the QNK-TT program [14] for a typical horizontal conveyor for the underground mining of copper ore, whereø133 × 465 idlers with cast iron hubs are used.

The data adopted for the calculations are presented in Table 3.

**Table 3.** Parameters of an exemplary conveyor.

|  |  | Copper Ore |
|---|---|---|
| Material Conveyed | Maximum Lump Size $d_{max}$ | 0.250 m |
|  | Percentage of Clumps $p_b$ | 20% |
|  | Surcharge Angle $\rho$ | 15° |
|  | Material Density $\gamma$ | 1950 kg/m$^3$ |
| Conveyor parameters | Conveyor design capacity $Q_n$ | 2990 t/h |
|  | Belt speed v | 2.50 m/s |
|  | Conveyor length L | 1000 m |
|  | Inclination angle $\delta$ | 0.00° |
|  | Vertical radius $R_v$ | 0 m |
|  | Idler spacing $l_g$ | 0.80 m |
|  | Troughing angle $\beta$ | 35° |
| Belt | Belt width B | 1200 mm |
|  | Belt mass | mt = 31.18 kg/m |
|  |  | mtj = 25.98 kg/m2 |
| Idlers | Description | G-H-133x465/6305-INTERkraz |
|  | Mass of idler | mk = 9.79 kg |
|  | Mass of idler rotating parts | m_rot = 7.43 kg |

Comparative calculations were carried out taking into account the measured values of rolling resistances for idlers 3 and 12. Based on the tests, the average rolling resistance for

idler 3 was assumed to be 5.1 N, and for idler 12 it was 1.9 N. For this purpose, the main resistances of the conveyor belt were considered as follows:

- Wk—rolling resistances of idlers,
- We—resistance to indentation of the idler casing into the belt covers,
- Wr—belt bending resistance on idler sets,
- Wf—resistance related to the heaving of output,
- Ws—resistance of belt sliding on the idlers.

From the point of view of the energy effect evaluation, after using certain idlers, an important factor is the share of idler rolling resistance Wk in the main resistance of the conveyor. The application of idlers with high-quality Y bearings, with C4 clearance changed the structure of the main resistances; the share of idler rolling resistances decreased from 18% to 8% of the main resistances of the conveyor in relation to the idler with X bearings with C3 clearance.

Further analysis of the structure of the conveyor's resistance to motion allowed for drawing a bar graph of the components of the conveyor's main resistance so as to compare the effect of using idlers with high-quality bearings. This graph is shown in Figure 13.

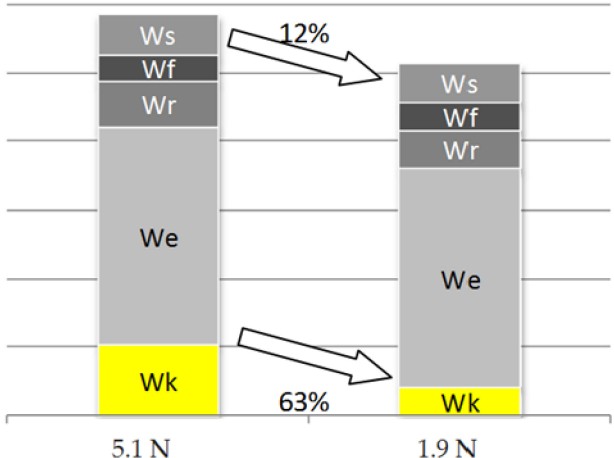

**Figure 13.** Comparison of the effect of using idler 3 (**left**) and 12 (**right**).

The diagram in Figure 13 shows that the use of idlers, as concluded in this research, causes a decrease in idler rolling resistance by 63%, which in turn contributes to a reduction in the main resistance and, at the same time, a decrease in the energy consumption of the conveyor by 12%.

The above analysis was performed for idlers with the highest and the lowest values of rolling resistance among the tested ones. These idlers differed in both the bearing manufacturer and the bearing clearance. In the next stage of analysis, it was checked how these features, considered separately, reduced the main resistance of the conveyor.

In the diagram in Figure 14, the effects of using X and Y bearings in the idler have been compared.

The compared idlers were equipped with C4 clearance bearings. Idlers 7 and 10 had L1 axial play, idlers 8 and 11 had L2, while idlers 9 and 12 had L3 axial play. A comparison of the characteristics of the manufacturer (quality) of the bearing have shown that better-quality bearings can reduce the conveyor motion resistance by a maximum of 10%.

The diagram in Figure 15 illustrates the effect of using Y bearings with C3 and C4 clearance. Idlers 4 and 10 had L1 axial play, idlers 5 and 11 had L2, whereas idlers 6 and 12 had L3 axial play.

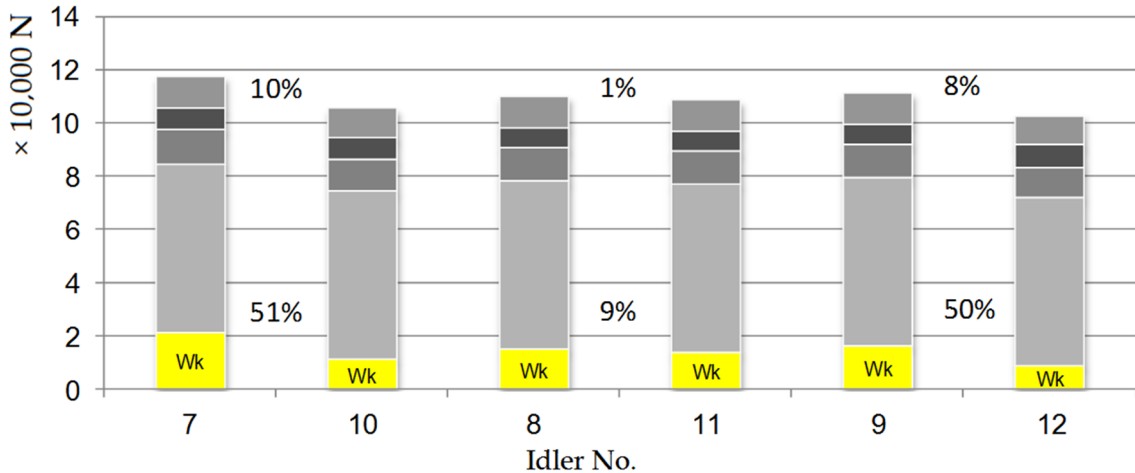

**Figure 14.** Comparison of the effect of using X bearings (idlers 7, 8 and 9) with Y bearings (idlers 10, 11 and 12).

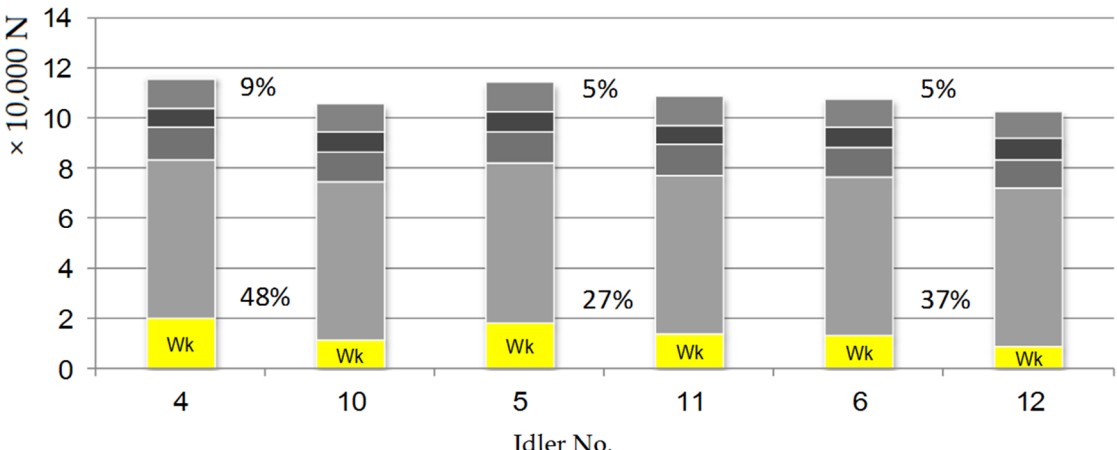

**Figure 15.** Comparison of the effect of using bearings with C3 clearance (idlers 4, 5 and 6) and bearings with C4 clearance (idlers 10, 11 and 12).

The application of bearings of a given manufacturer with C4 bearing clearance compared to C3 results in reducing the conveyor motion resistance by 5 to 9%.

## 5. Conclusions

The research carried out by the authors of this study was aimed at determining what energy effects can be brought about by making relatively simple changes in the idler structure. An important aspect of the investigations into idler rolling resistance was conducting them under loading conditions that occur during operation. Therefore, idlers were subjected to radial and axial loads.

The presented results of the laboratory tests show that the condition of idler load has a significant impact on its motion resistance. Idlers loaded with both radial and axial forces can have more than 2.5-times greater rolling resistance than those loaded only with the radial force, which is important for the design of belt conveyors.

This research has also revealed that appropriate selection of, in particular, idler bearings, enables obtaining measurable effects in the form of lower energy consumption of the belt conveyor. The calculations based on the research results in the QNK-TT program have demonstrated that the power consumption by the drive of an exemplary conveyor, typically used in copper ore mines, can be reduced by approximately 10% by using bearings with C4 instead of C3 clearance in idlers. Considering the scale of application of belt transport in the global raw materials industry, this is an important piece of information.

The graph presented in Figure 16 shows the effect of idler rolling resistance on the power consumption of an exemplary belt conveyor. It indicates that lowering the rolling resistance of each idler by 1N reduces the energy consumption of the loaded conveyor by 4%.

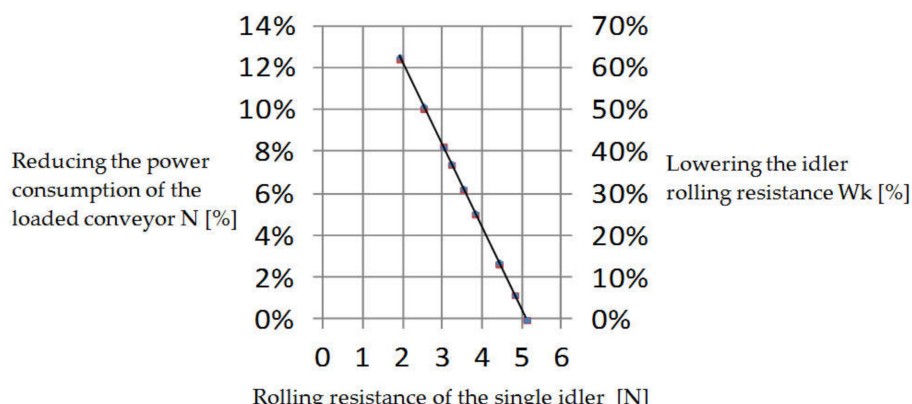

**Figure 16.** Dependence of the conveyor drive power on idler rolling resistance.

In terms of their structure, idlers are relatively simple components of the belt conveyor. However, the number of idlers installed in the conveyors makes them a significant source of resistance to motion, and, in consequence, the power consumption of the conveyor drive. At the same time, idlers are still potential reserves for reducing the energy consumption of belt transport [15,16]. Improving the energy efficiency of the belt conveyors is an important field for researchers and engineers [17,18], and the results presented in this publication provide a significant contribution to that field. The authors of this study have demonstrated that simple, low-cost changes to the idler structure can result in noticeable effects increasing the efficiency of belt transport. Safe and efficiently implemented belt transport processes are an important aspect of the sustainable development of the raw materials industry.

**Author Contributions:** Conceptualization, P.K. (Piotr Kulinowski) and P.K. (Piotr Kasza); investigation, P.K. (Piotr Kasza) and J.Z.; supervision, P.K. (Piotr Kasza); visualization, P.K. (Piotr Kulinowski) and J.Z.; writing—original draft, P.K. (Piotr Kasza); writing—review and editing, P.K. (Piotr Kulinowski), J.Z.; formal analysis, P.K. (Piotr Kulinowski). All authors have read and agreed to the published version of the manuscript.

**Funding:** This research received no external funding.

**Institutional Review Board Statement:** Not applicable.

**Informed Consent Statement:** Not applicable.

**Data Availability Statement:** Data sharing not applicable.

**Conflicts of Interest:** The authors declare no conflict of interest.

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
