# Peer review of "Influence of Design Parameters of Idler Bearing Units on the Energy Consumption of a Belt Conveyor"

_sustainability, doi:10.3390/su13010437_

Round 1
Reviewer 1 Report
1. The author should change paragraph lengths in Section 1 so that Table 1 fits on one page. As presented to me, Table 1 is split over 2 pages. This needs to be corrected.
2. Ditto Section 3, Table 2 - Fix split Table and make Table 2 reside all together on a page.
3. Author should add a comment up front in the Abstract that says "the test rig measures idler bearing losses and rolling drag, not belt indentation rolling resistance".
4. Author should make it clear / ADD that ;
"the test rig only applies to testing individual idler rolls, and not 2, 3 or 5 idler roll sets commonly used to support the carrying trough of actual belt conveyors. Since the test rig measures axial load resistance, its application to flat idler rolls in a 35 degree (or any other angled trough) troughed sets will need to be considered and adjusted in the results, so that a designer may use the data in a real situation. A curve is showing how axial rolling resistance due to bearing losses varies with trough angle for wing idler roll would be beneficial to designer engineers".
5. Reference 7 - Craig Wheeler : The words in the paper title run together and need to be edited to separate these words in the reference.
Author Response
Point 1: The author should change paragraph lengths in Section 1 so that Table 1 fits on one page. As presented to me, Table 1 is split over 2 pages. This needs to be corrected. 

Response 1: The author agree with the reviewer. Will be corrected.
Point 2: Ditto Section 3, Table 2 - Fix split Table and make Table 2 reside all together on a page.
Response 2: The author agree with the reviewer. Will be corrected.
Point 3: Author should add a comment up front in the Abstract that says "the test rig measures idler bearing losses and rolling drag, not belt indentation rolling resistance".
Response 3: A comment added in the Abstract.
Point 4: Author should make it clear / ADD that ;
"the test rig only applies to testing individual idler rolls, and not 2, 3 or 5 idler roll sets commonly used to support the carrying trough of actual belt conveyors. Since the test rig measures axial load resistance, its application to flat idler rolls in a 35 degree (or any other angled trough) troughed sets will need to be considered and adjusted in the results, so that a designer may use the data in a real situation. A curve is showing how axial rolling resistance due to bearing losses varies with trough angle for wing idler roll would be beneficial to designer engineers".
Response 4: Added in section 2.
Point 5: Reference 7 - Craig Wheeler : The words in the paper title run together and need to be edited to separate these words in the reference.
Response 5: The author agree with the reviewer. Will be corrected.
Reviewer 2 Report
The introduction presents the issue of the study of resistance to the movement of idlers. The literature cited is scarce. It would be a good idea to add some more international positions. Authors undertook research work to better understand the real working conditions of the idlers (especially motion resistances). They indicate that this topic has not been sufficiently described in the available literature. However another goal of the research would be to provide the contribution necessary to increase the energy efficiency of the belt conveyors. It would be worthwhile to mention this issue, e.g. in the introduction.
The description of the test rig is clear. It would be a good idea to clearly explain how the charts are constructed. The frequency of force recording is 1kHz, the resulting lines in the graph are smooth. It was probably necessary to use a moving average to get such result.
The discussion is done in an accurate way. The conclusion set out how reducing the resistances can increase the energy efficiency of the belt conveyor transport. It is worth describing what are the possibilities to implement the knowledge resulting from research.
Author Response
Point 1: The introduction presents the issue of the study of resistance to the movement of idlers. The literature cited is scarce. It would be a good idea to add some more international positions. Authors undertook research work to better understand the real working conditions of the idlers (especially motion resistances). They indicate that this topic has not been sufficiently described in the available literature. However another goal of the research would be to provide the contribution necessary to increase the energy efficiency of the belt conveyors. It would be worthwhile to mention this issue, e.g. in the introduction.
The description of the test rig is clear. It would be a good idea to clearly explain how the charts are constructed. The frequency of force recording is 1kHz, the resulting lines in the graph are smooth. It was probably necessary to use a moving average to get such result.
The discussion is done in an accurate way. The conclusion set out how reducing the resistances can increase the energy efficiency of the belt conveyor transport. It is worth describing what are the possibilities to implement the knowledge resulting from research.
Response 1: The Author added number of four international positions, as reviewer suggested. They were added in the introduction (section 1, position 12 and 13) and in the conclusions (section 5 , position 17 and 18).
The explanation how the charts are constructed is on page 5. The Author used a moving average to get smooth lines on charts.
The author described in section 5 that the practical implementation of the results of this work can be done by installing in rollers high quality bearings with specific clearance with taking into account roller's axial play.